# Mapping of Genomic Vulnerabilities in the Post-Translational Ubiquitination, SUMOylation and Neddylation Machinery in Breast Cancer

**DOI:** 10.3390/cancers13040833

**Published:** 2021-02-17

**Authors:** Jesús Fuentes-Antrás, Ana Lucía Alcaraz-Sanabria, Esther Cabañas Morafraile, María del Mar Noblejas-López, Eva María Galán-Moya, Mariona Baliu-Pique, Igor López-Cade, Vanesa García-Barberán, Pedro Pérez-Segura, Aránzazu Manzano, Atanasio Pandiella, Balázs Győrffy, Alberto Ocaña

**Affiliations:** 1Experimental Therapeutics Unit, Hospital Clínico Universitario San Carlos (HCSC), Instituto de Investigación Sanitaria San Carlos (IdISSC) and Centro de Investigación Biomédica en Red en Oncología (CIBERONC), 28040 Madrid, Spain; esther.cabanas@salud.madrid.org (E.C.M.); maria.baliu.pique@sergas.es (M.B.-P.); pedro.perez@salud.madrid.org (P.P.-S.); aranzazu.manzano@salud.madrid.org (A.M.); 2Translational Oncology Laboratory, Centro Regional de Investigaciones Biomédicas, Castilla-La Mancha University (CRIB-UCLM), 02008 Albacete, Spain; analucia.alcaraz@uclm.es (A.L.A.-S.); mariadelmar.noblejas@uclm.es (M.d.M.N.-L.); evamaria.galan@uclm.es (E.M.G.-M.); 3Molecular Oncology Laboratory, Instituto de Investigación Sanitaria San Carlos (IdISCC), 28040 Madrid, Spain; ilopez.7@alumni.unav.es (I.L.-C.); vanesa.garciabar@salud.madrid.org (V.G.-B.); 4Instituto de Biología Molecular y Celular del Cáncer (IBMCC), Consejo Superior de Investigaciones Científicas (CSIC-IBSAL) and Centro de Investigación Biomédica en Red en Oncología (CIBERONC), 37007 Salamanca, Spain; atanasio@usal.es; 5Department of Bioinformatics, Semmelweis University, H-1094 Budapest, Hungary; gyorffy.balazs@ttk.hu; 62nd Department of Pediatrics, Semmelweis University, H-1094 Budapest, Hungary; 7TTK Cancer Biomarker Research Group, Institute of Enzymology, H-1117 Budapest, Hungary

**Keywords:** post-translational modification, ubiquitination, SUMOylation, neddylation, breast cancer, biomarkers, prognosis

## Abstract

**Simple Summary:**

Breast cancer is a major cause of death worldwide and remains incurable in advanced stages. The dysregulation of the post-translational machinery has been found to underlie tumorigenesis and drug resistance in preclinical models but has only recently led to early trials in cancer patients. We performed an in silico analysis of the most common genomic alterations occurring in ubiquitination and ubiquitin-like SUMOylation and neddylation using data from publicly available repositories and with the aim of identifying those with prognostic and predictive value and those exploitable for therapeutic intervention. Clinical and statistical criteria were used to sort out the best candidates and the results were validated in independent datasets. *UBE2T*, *UBE2C*, and *BIRC5* amplifications predicted a worse survival and poor response to therapy across different intrinsic subtypes of breast cancer. Mutated *USP9X* and *USP7* also conferred detrimental outcome. Leveraging these molecular vulnerabilities as biomarkers or drug targets could benefit breast cancer patients.

**Abstract:**

The dysregulation of post-translational modifications (PTM) transversally impacts cancer hallmarks and constitutes an appealing vulnerability for drug development. In breast cancer there is growing preclinical evidence of the role of ubiquitin and ubiquitin-like SUMO and Nedd8 peptide conjugation to the proteome in tumorigenesis and drug resistance, particularly through their interplay with estrogen receptor signaling and DNA repair. Herein we explored genomic alterations in these processes using RNA-seq and mutation data from TCGA and METABRIC datasets, and analyzed them using a bioinformatic pipeline in search of those with prognostic and predictive capability which could qualify as subjects of drug research. Amplification of *UBE2T*, *UBE2C*, and *BIRC5* conferred a worse prognosis in luminal A/B and basal-like tumors, luminal A/B tumors, and luminal A tumors, respectively. Higher *UBE2T* expression levels were predictive of a lower rate of pathological complete response in triple negative breast cancer patients following neoadjuvant chemotherapy, whereas *UBE2C* and *BIRC5* expression was higher in luminal A patients with tumor relapse within 5 years of endocrine therapy or chemotherapy. The transcriptomic signatures of *USP9X* and *USP7* gene mutations also conferred worse prognosis in luminal A, HER2-enriched, and basal-like tumors, and in luminal A tumors, respectively. In conclusion, we identified and characterized the clinical value of a group of genomic alterations in ubiquitination, SUMOylation, and neddylation enzymes, with potential for drug development in breast cancer.

## 1. Introduction

Despite the continuous improvement of diagnostic and therapeutic strategies, virtually all patients with advanced breast cancer (BC) will die from this disease [1]. In recent years, large-scale genomic studies have generated an overwhelming pool of data that reflects the complexity and heterogeneity of BC [2,3,4,5,6]. From a clinical standpoint, the progressive genomic characterization has helped envision potential druggable targets based on genomic vulnerabilities, some of which have reached clinical implementation at different levels as exemplified by the growing importance of the ESMO Scale of Clinical Actionability in clinical trial design [7,8,9]. However, a large number of molecular alterations remain unexplored, including many with potential for medical development.

Posttranslational modifications (PTM), defined as the covalent and generally enzymatic modification of proteins following biosynthesis, are progressively gaining momentum in cancer research as they play a transversal role in many biological processes found aberrant in carcinogenesis, including protein diversification, degradation and recycling, gene regulation, oncogenic signaling (i.e., p53, NF-κB, TGF-β pathways) and DNA damage repair [10,11]. These modifications encompass phosphorylation, acetylation, glycosylation, and ubiquitination, among many others, and are subject to exquisite regulatory mechanisms often altered in cancer. Considering their critical roles and their widespread dysregulation in cancer, there is a growing interest in developing therapeutic approaches capable of specifically exploiting the ubiquitin and ubiquitin-like pathways (namely, SUMO and Nedd8) in solid tumors. From a molecular perspective, ubiquitin is a small 76-amino-acids protein that can be conjugated to target proteins by three types of enzymes including ubiquitin-activating enzymes (E1s), ubiquitin-conjugating enzymes (E2s), and ubiquitin ligases (E3s) [12]. Small ubiquitin-like modifier (SUMO) and Nedd8 are ubiquitin-like proteins with the same enzymatic structural design as ubiquitin modification, requiring E1-activating, E2-conjugating, and E3-ligating enzymes. Although these three PTMs (SUMOylation, ubiquitination, and neddylation; collectively termed SUN) exert pleiotropic functions in common processes such as cell signaling, inflammatory responses, and DNA damage repair, ubiquitination plays a primary role in the degradation and recycling of the proteome, while SUMOylation and neddylation participate in protein stabilization, transcriptional regulation, epithelial-mesenchymal transformation, and apoptosis [10,11,13,14,15]. 

Drug development based on molecular vulnerabilities occurring in the SUN machinery has only recently emerged, although the ubiquitin-conjugation-associated proteasome pathway is already targeted by FDA approved drugs (i.e., Bortezomib) used in malignancies such as multiple myeloma [16]. A comprehensive overview of preclinical and ongoing human trials targeting multiple SUN components was recently published by Gâtel et al., with most research focused on hematologic malignancies and in early phases of development [11]. In BC models, the targeted inhibition of MDM2, an E3-ubiquitin-ligase, has shown promising therapeutic effects either through its autoubiquitination and proteasomal degradation by SP-141 or by blocking the p53-MDM2 interaction by WK298 and SJ-172550 [17,18]. Interestingly, proteolysis targeting chimeras (i.e. PROTACs), developed to trigger ubiquitination and subsequent degradation of specific proteins, have recently proved to exert superior tumor growth inhibition compared to fulvestrant by targeting the estrogen receptor (ER) in a patient-derived xenograft model from a patient harboring a mutation in the ER, and a similar PROTAC is currently being tested in ER positive HER2 negative BC patients in a phase I trial (NCT04072952) [19]. Furthermore, these basic mechanisms of proteostasis have not been studied as potential biomarkers of drug efficacy or prognosis in BC. 

In this article, we explored genomic alterations in the SUN machinery occurring in BC. With a clinical scope and in search of potential actionability, we focused on amplifications with a prognostic correlation and leading to increased levels of enzymes, thereby filtering for intuitive targets for drug development. By interrogating multiple large publicly accessible databases and analytical webs for cancer omics, we found that gene amplifications are the most frequent genomic alteration of the SUN components. An association with a worse clinical outcome was identified for *SUMO2*, *TCEB2*, *BIRC5*, *UBE2T*, *DERL1*, *PRKDC*, and *UBE2C* gene amplifications. We further described that the amplification of *UBE2T*, *UBE2C*, and *BIRC5*, remain significant and clinically relevant predictors of a worse outcome in luminal A/B and basal-like, luminal A/B, and luminal A subtypes, respectively. This prognostic signature also showed predictive potential after stratifying by treatments administered in the (neo)adjuvant setting. Finally, although less common, mutations in *USP9X* and *USP7* also predicted a worse outcome in different intrinsic subtypes. In summary, we report the potential prognostic and predictive capability of genomic alterations in SUN enzymes and provide a rationale for future drug development.

## 2. Results

### 2.1. Copy Number Alterations in SUMOylation, Ubiquitination, and Neddylation in Breast Cancer

Reactome, a curated database of biological pathway knowledge, was sequentially interrogated for genes involved in the metabolism of proteins, post-translational protein modifications, and specifically the processes of SUMOylation, ubiquitination, and neddylation. A total of 113 genes were identified (Figure 1a). Of them nine, 77, and 28 genes were involved in SUMOylation, ubiquitination, and neddylation, respectively (Figure 1b). Only one gene (*UCHL3*) belonged to two groups (neddylation and ubiquitination). In order to limit redundance, we confirmed that such functional characterization was recapitulated in the EnrichR database (Figure 1c). We explored the TCGA (Cell 2015) and METABRIC datasets in the cBioPortal online platform to identify frequent alterations in these genes in BC irrespective of their intrinsic subtype (Figure 1d; Appendix A) [20,21]. Tumor mutational and copy number alteration (CNA) data was available in 2989 of patients combining both datasets. Gene amplifications were by far the most frequent alterations occurring in SUN processes, with the highest proportion observed in the *UBE2T* gene, in 23.46% of patients. In this work, we focused on the study of these alterations since genomic amplifications, and subsequently increased levels of enzymes, underlie many oncogenic pathways and resistance mechanisms and thereby provide intuitive targets for drug development. 

### 2.2. Filtering of Gene Amplifications in Search of Potentially Relevant Clinical Targets

Next, we explored the concordance of gene amplifications across the two datasets to identify those alterations homogenously distributed in the whole population available. We performed a regression analysis and set a 2% alteration frequency threshold to curate for potential clinical relevance (Figure 2a). The selected genes were further analyzed according to their amplification frequency across the different intrinsic subtypes of BC, with no evident clustering of amplifications in any of them (Figure 2b).

To evaluate whether gene amplifications truly impacted mRNA levels, we first interrogated the GEPIA2 web tool [22,23]. Significantly increased mRNA levels were found in breast tumors compared to healthy controls for *UBE2T*, *UBE2C*, *BIRC5*, *TCEB2* (Figure 3). Although non-significant, most of the other genes consistently showed increased mRNA levels in tumor samples. Further, the bc-GenExMiner 3.0 and UALCAN databases were used to study how the expression of those genes with significantly increased mRNA levels distributed across the different intrinsic subtypes and tumor stages [23,24]. Remarkably, mRNA levels of *UBE2T*, *UBE2C,* and *BIRC5,* were found significantly higher in basal-like tumors compared to luminal A tumors, whereas *TCEB2* mRNA levels were found higher in luminal A tumors compared to basal-like tumors, although this difference was slight. By analyzing individual patient data from the TCGA database, we confirmed that patients with gene amplifications had higher transcriptional levels across the different intrinsic subtypes (Figure 3). For all genes, expression tended to be highest in stage IV samples and lowest in stage I (Appendix A). Similarly, protein levels were highest in basal-like tumors and lowest in luminal tumors, with the exception of *TCEB2*, thus pointing to a positive correlation between the protein burden and a more dedifferentiated phenotype (Appendix A). Also of note, a significantly lower promoter methylation level was found in the *UBE2T* gene in BC samples with respect to normal tissues, which is consistent with an enhanced gene expression. Conversely, *BIRC5* showed a higher promoter methylation level in tumors (Appendix A).

### 2.3. Amplification of SUN Genes Predicts an Unfavorable Outcome in Breast Tumors

The KM plotter tool was used to explore the prognostic role of each gene amplification occurring in more than 2% of patients in both TCGA and METABRIC datasets [25]. High transcriptional levels of *SUMO2*, *TCEB2*, *BIRC5*, *UBE2T*, *DERL1*, *PRKDC,* and *UBE2C*, were associated with a shorter relapse-free survival (RFS) and their Kaplan-Meier curves are displayed in Figure 4a. Further, we validated these results by interrogating the METABRIC and TCGA datasets and found that *BIRC5*, *UBE2T*, and *UBE2C* remained significantly linked to a worse RFS with a false discovery rate (FDR) <5% for almost all associations (Appendix A; clinicopathological features of patients from TCGA and METABRIC datasets with survival data are summarized in Appendix A). Stratification by nodal involvement and histological grade did not substantially modify the association with outcome, although most analysis presented an FDR >10%, probably due to the limited population sample in each group (Appendix A). Only *BIRC5*, *UBE2T* and *UBE2C* remained significantly linked to a worse RFS with an FDR of 1% in the subgroup of node positive patients. Interestingly, most of these amplifications were co-occurrent within the same tumors, displaying a strong statistical association (Figure 4b), and yet cytogenetic mapping showed that only *BIRC5* and *SUMO2*, and *DERL1* and *PRKDC*, are located in the same chromosomes and arms (17q25.3 and 17q25.1, and 8q24.13 and 8q11.21, respectively). The STRING database was used to map their interaction network revealing a significant enrichment *p*-value and low-to-moderate local clustering coefficient, defined as the likelihood of the connection (Figure 4c). 

### 2.4. Amplification of UBE2T, UBE2C, and BIRC5 Genes Is Associated to a Worse Prognosis in Luminal and Basal-Like Breast Tumors

In order to further filter for potential clinical relevance, we next investigated the prognostic implication of those amplified genes across intrinsic subtypes. Despite showing a consistent prognostic capability, most of them did not pass a restrictive screening excluding those not reaching statistical significance, with an FDR over 5%, and/or with low clinical impact (defined as a HR of RFS < 1.5) (Figure 5a). High levels of *UBE2T* predicted a poorer RFS in luminal A/B and basal-like tumors, whereas *UBE2C* and *BIRC5* associated with a worse outcome in luminal A/B and luminal A tumors, respectively (Figure 5b). Validation in the METABRIC and TCGA datasets showed consistent results, with the exception of a loss of statistical significance of *UBE2T* in basal-like tumors and overall higher FDRs as a result of the reduced population available (Appendix A). 

### 2.5. Amplification of UBE2T, UBE2C, and BIRC5 Genes Can Predict Response to (neo)adjuvant Therapy in Luminal A and Basal-Like Tumors

The ROCplot online tool was used to evaluate whether higher transcriptional levels of *UBE2T*, *UBE2C*, and *BIRC5*, could qualify as putative biomarkers of response to (neo)adjuvant therapy in BC [26]. In the neoadjuvant setting, *UBE2T* performed substantially well as a surrogate of a worse rate of pathological complete response (pCR) after chemotherapy in TNBC (AUC 0.629, chi-square *p* = 0.028) (Figure 6a). In the adjuvant setting, a higher expression of *UBE2C* and *BIRC5* predicted a poorer response in luminal A patients treated with chemotherapy (AUC 0.796, chi-square *p* = 8.9 × 10^−5^, and AUC 0.737, chi-square *p* = 0.017, respectively), although the limited number of evaluable patients calls for a cautious interpretation and further validation (Figure 6b). A higher expression of *UBE2C* and *BIRC5* also significantly, although slightly, correlated with a shorter RFS in luminal A patients receiving adjuvant hormone therapy (AUC 0.588, chi-square *p* = 0.0093, and AUC 0.577, chi-square *p* = 0.024, respectively) (Figure 6c). A control analysis was performed assessing the expression of *ESR1*, a clinically established biomarker predictive of response to endocrine therapy. A combined signature of *UBE2C* and *BIRC5* was statistically significant in both scenarios but did not increase the predictive capability.

### 2.6. Mapping of Mutations Occurring in SUN Genes and Prognostic Impact

Although mutations in SUN components were not common, we explored their landscape in TCGA to further characterize their potential clinical interest (Figure 7a). Mutations occurring in more than 0.5% of patients were compared across intrinsic subtypes and their prognostic role was evaluated using the Genotype2Outcome tool (Figure 7b,c) [27]. The transcriptional signature of each gene mutation was calculated as described in Material and Methods. The signature associated to *USP9X* predicted a worse overall survival in luminal A, HER2-enriched, and basal-like tumors, while that linked to *USP7* predicted a worse outcome in luminal A tumors (Figure 7c). Figure 7d shows the top genes ranked by *p*-value for *USP9X* and *USP7*, respectively. The distribution of up- and downregulated genes contributing to the signature and the individual HR of RFS of the top ones ranked by *p*-value are displayed in Appendix A. Using the same dataset, we explored the type of mutation described in these genes (Appendix A). A total of 13 patients had *USP9X* mutations, from which 9 were missense and 5 were truncating, whereas 6 patients had *USP7* mutations, from which 5 were missense and 1 was truncating. No hotspots were observed. Only a single patient presented mutation co-occurrence, a trend which was not significant (Log2 OR >3, q = 0.092). Online tools predicting the functional effects of these mutations (i.e., SIFT, PolyPhen, MutationTester, FATHMM-MKL) agreed in the deleterious impact of 2 missense mutations in *USP7* (c.3011G>A and c.1223G>A) and of 5 missense mutations in *USP9X* (c283G>C, c.1969C>T, c2225G>A, c.3894G>C, and c.6586C>G), none of which are archived in the ClinVar database. Although not described in the tools above mentioned, a total of 3 frameshift mutations occurring early in *USP9X* (c.273_274del, c.2084_2090del, and c.4104_4105del) are likely to be deleterious, as well as 1 stop mutation occurring in each *USP9X* (c.1861C>T) and *USP9X* (c.2248C>T). A single mutation of a donor splice site (c.5331+1G>A) was also identified and predicted to result in the loss of function of that site using the MaxEntScan tool, but its impact on *USP9X* activity remains unknown.

### 2.7. Exploring Genomic Vulnerabilities in a Large Panel of Breast Cancer Cell Lines 

Finally, we searched for putative in vitro models that could help characterize the molecular underpinnings of the genomic alterations identified from patient data. To this end, we interrogated the Cancer Dependency Map (DepMap) portal, the Catalogue of Somatic Mutations in Cancer (COSMIC), and at the Cancer Cell Line Encyclopedia (CCLE), as described in Materials and Methods [28,29,30,31]. In contrast to the high prevalence of *UBE2T*, *UBE2C*, and *BIRC5* amplifications in BC patients, only 2/49 cell lines (JIMT-1 and HCC1569) harbored *UBE2C* amplifications and 3/49 cell lines (HCC1143, OCUB-M, and MDA-MB-361) harbored *BIRC5* amplifications (Figure 8a; Appendix A). The copy number of these genes positively correlated with mRNA levels, and transcript levels were strongly associated with Ki67 expression, which is consistent with a more proliferative phenotype and a poorer prognosis (Figure 8b,c). Expression of *UBE2T*, *UBE2C* and *BIRC5* distributed across intrinsic subtypes in a similar fashion to BC patients, with a tendency to be highest in basal-like cells and lowest in luminal cells. However, this trend lacked statistical significance, likely due to the limited sample size (Figure 8d). More cell lines presented mutations in *USP9X* (8/49) and *USP7* (5/49), although an unexpected high frequency of *USP9X* mutations in luminal cell lines was found along with a poor overall concordance with the mutational profile of patients. The predictive capability of the set of gene amplifications was explored using data from comprehensive online repositories (see Materials and Methods). No robust associations were observed, with varying trends among databases and reduced sample sizes as a result of our effort to stratify cell lines according to their intrinsic subtypes to recapitulate the actual clinical setting (Appendix A).

## 3. Discussion

In the present article we describe common genomic alterations in SUN genes in BC. By means of a comprehensive bioinformatic pipeline, the main goal of this work was to identify those genes linked to a worse prognosis and that therefore could be explored in the future as druggable targets and/or predictors of response to treatment.

By using TCGA and METABRIC datasets (combined *n* = 2989), we observed that amplifications are by far the most frequent genomic alteration occurring in SUN genes. We further identified five, 15, and six genes that presented a frequency of gene amplifications of more than 2% in SUMOylation, ubiquitination, and neddylation pathways, respectively. We refined our search by selecting only those with more than 2% of gene amplifications in both databases individually, thus reducing the list to four, 11 and four for the abovementioned processes, respectively. The filtered genes were then stratified across the different intrinsic subtypes. The mRNA levels of amplified genes tended to be higher in breast tumors than in the healthy state, although this difference reached statistical significance only for *UBE2T*, *BIRC5*, *UBE2C*, and *TCEB2*. Also, with the exception of *TCEB2*, gene expression and protein levels tended to be highest in basal-like tumors and lowest in luminal A tumors. Overall, gene amplification led to higher transcriptional levels across subtypes. We may acknowledge that although the frequency of genomic alterations is useful to uncover potential molecular vulnerabilities, the magnitude of the amplification of each gene is better represented by the amount of mRNA levels, and thus the prognostic and predictive correlations shown in our work are based on expression levels. Identification of tumors harboring amplifications is easier to be implemented in the clinical setting as has been the case in BC for *HER2*. Of note, many genes that are not frequently amplified but may have high expression rates and be of prognostic interest in BC are not discussed in our analysis. From the set of genes, *SUMO2*, *TCEB2*, *BIRC5*, *UBE2T*, *DERL1*, *PRKDC*, and *UBE2C* showed a significant association with a worse RFS. Co-occurrence of gene amplifications was commonly found along with a robust functional network. Relevant associations (HR of RFS >1.5) remained statistically significant and with an FDR ≤5% for *UBE2T*, *UBE2C*, and *BIRC5*, in luminal A/B and basal-like tumors, luminal A/B tumors, and luminal A tumors, respectively. Most of these results were recapitulated in the METABRIC and TCGA databases as validation. When exploring the predictive capability of these amplified genes, *UBE2T* expression levels were found significantly increased in TNBC patients not achieving pCR after neoadjuvant chemotherapy, whereas *UBE2C* and *BIRC5* expression were higher in luminal A patients with tumor relapse within 5 years of endocrine therapy or chemotherapy.

Finally, we sought to characterize the mutational landscape and prognostic implications of the SUN components, and found that mutated *USP9X* and *USP7* genes, analyzed through their surrogate transcriptomic signatures, were linked to a worse prognosis in luminal A, HER2-enriched, and basal-like tumors, and in luminal A tumors, respectively. Main findings and potential therapeutic approaches are outlined in Table 1.

Establishing a bioinformatic pipeline for omic analysis has progressively become a critical need for the discovery of therapeutic targets in cancer. These tools, guided by clinical experience, can provide intuitive access to a growing pool of data and generate fast and robust answers to experiments without inefficiently compromising patients. We showed that ubiquitin-conjugating enzymes E2T and E2C, and baculoviral IAP repeat containing 5 (*BIRC5*; also known as survivin) are amplified in BC and associated to a worse prognosis across subtypes and stages. Importantly, genomic amplifications correlated with increased mRNA levels. *UBE2T* and *UBE2C* have been previously associated with cancer progression and poor outcome in several solid tumors, although genomic evidence in BC is recent and limited [32,33,34,35,36]. Consistently, in vitro functional experiments in ER positive HER2 negative BC cells showed that *UBE2C* expression is a tumorigenic factor, that it is regulated by estrogen through direct binding to the *UBE2C* promoter region, and that its overexpression leads to estrogen-independent growth [37]. Moreover, *UBE2C* depletion markedly increased the cytotoxicity of letrozole, tamoxifen, doxorubicin, and the sensitivity to radiation therapy in multiple BC cell lines [37,38]. In this regard, we showed that higher *UBE2C* mRNA levels are not only a prognostic factor but also predictive of a worse response to adjuvant endocrine therapy and chemotherapy in patients with luminal A tumors. Less explored so far, our group recently reported first evidence that *UBE2T* is amplified in BC and non-small cell lung carcinomas, in which it was related to a detrimental outcome [36]. In the present work we describe that *UBE2T* predicts a worse outcome in luminal A/B and basal like tumors, and also a poorer response to neoadjuvant chemotherapy in TNBC. As a hint for its molecular rationale, *UBE2T* overexpression has been found to facilitate cell cycle progression and to avoid DNA repair by degrading key regulators of both functions such as p21 or BRCA1 in vitro and in vivo in BC [39]. This could also pave the way to exploring the benefit of PARP inhibitors in patients harboring somatic *UBE2T* overexpression independent of their BRCA status.

In addition to ubiquitin, the ubiquitin superfamily also contains ubiquitin-like proteins, including Nedd8 and SUMO, which do not only have sequence homology and structural similarity to ubiquitin but also use a similar enzymatic cascade to modify their substrate proteins [53]. We reported that *BIRC5* amplification is a biomarker of a worse prognosis in luminal A patients, and also predicts a worse response to adjuvant endocrine therapy or chemotherapy in these patients. These findings are consistent with a recent *in silico* analysis by Dai et al. and with the observation by Hamy et al. that *BIRC5* predicts a worse RFS in stage II/III BC patients of all intrinsic subtypes who do not achieve a pCR after neoadjuvant chemotherapy [40,41]. Surprisingly, the promoter methylation level of *BIRC5* was found to be increased in tumor samples, which appears to go against its expression pattern. Although this finding has not been clearly explained in BC, previous observations in endometrial cancer have found that the hypermethylation of the *BIRC5* promoter blocks the binding of p53, a repressor of *BIRC5* gene transcription, to its promoter region, thus increasing its expression [42,43]. Also interestingly, *BIRC5* is one of the genes included in the Oncotype DX^®^, Endopredict^®^ and Prosigna^®^ signatures. There is in vitro evidence that repressing *BIRC5* expression by siRNA could significantly inhibit proliferation and also induce a BRCAness phenotype with DNA double-strand breaks and a functional impairment of homologous recombination [44,45]. A comprehensive review by Li et al. has recently gathered together the advances in cancer therapeutics targeting BIRC5 [54]. 

We also found that mutations in *USP9X* and *USP7* have an ominous prognostic impact in luminal, HER2-enriched, and basal-like breast tumors. USP9X and USP7 are ubiquitin-specific proteases with roles somewhat antagonistic to UBE2T and UBE2C [55]. Consistently, inactivating missense and truncating mutations should recapitulate the effects of those amplifications. However, the molecular interplay underlying such clinical effects remains unclear. USP9X has been shown to stabilize BRCA1 and confer resistance to DNA-damaging agents, and yet its downregulation rendered BC cells resistant to tamoxifen, while its up-modulation has been found to promote centrosome amplification, chromosome instability, and higher histologic grades of BC, as well as to enhance Hippo pathway-dependent cell proliferation [46,47,48,49]. The role of USP7, in turn, appears to be more established in the regulation of DNA replication and mitosis progression by affecting the stability of Aurora-A kinase and Geminin [50,51]. Of note, USP7 physically interacts with the ERα, thereby mediating its deubiquitination and stabilization [52]. Altogether, this body of evidence underscores the potential utility of studying USP9X, USP7, and their associated transcriptomic signatures especially in luminal BC, although further functional characterization of specific mutations is required.

Given the potential interest of therapeutically exploiting this set of genomic vulnerabilities and the need to further describe their molecular foundations, we conducted a comprehensive *in silico* characterization of BC cell lines using mutational, transcriptional, and sensitivity data. Very few cell lines, from a panel including the most typically used and thoroughly described, presented the genomic alterations identified in patients. Their distribution across intrinsic subtypes also differed, and drug sensitivity analyses showed only slight tendencies that did not reach statistical significance. Molecular and pharmacological studies will be hindered in the absence of robust in vitro models recapitulating these gene alterations. However, the low frequency of the genomic alterations in cell lines has also been observed in other vulnerabilities (e.g., *MET* mutations in lung cancer or *BRCA1/2* in BC), and this fact has not impacted the clinical development of strategies against them. 

In sum, our data suggests that the amplification of *UBE2T*, *UBE2C*, and *BIRC5*, and inactivating mutations in *USP9X* and *USP7*, can predict a worse outcome in different intrinsic subtypes of BC. Comprehensive initiatives are currently focused on leveraging the prognostic implications of the ubiquitin and ubiquitin-like pathways in cancer. However, to the best of our knowledge, no targeted agents against UBE2T, UBE2C, BIRC5, or USP9X, either approved or in development, can be found in The Drug Gene Interaction Database or Genomics of Drug Sensitivity in Cancer database, while the only USP7 inhibitor, P22077, has not been tested in BC [56]. These findings are exploratory and require further validation in preclinical models and patient cohorts, particularly considering the influence of CDK4/6 inhibitors and newly approved targeted agents in BC, as well as further efforts to characterize the molecular and functional properties of the described alterations for drug design.

## 4. Materials and Methods 

### 4.1. Identification of SUN Genes Gene Ontology

Reactome (https://reactome.org/, accessed on: 8 January 2021), a publicly available relational database of signaling and metabolic molecules and their relations organized into biological pathways and processes, was used to sequentially narrow down genes involved the metabolism of proteins, post-translational modifications of proteins, and, finally, the processes of interest, that is, SUMOylation, ubiquitination, and neddylation. Further confirmation of the biological functions related to each gene set was obtained using the 2018 Molecular_function Gene Ontology Terms through the publicly available EnrichR online platform (https://maayanlab.cloud/Enrichr/, accessed on: 8 January 2021). 

### 4.2. Data Collection and Processing

We used data contained at cBioportal (www.cbioportal.org, accessed on: 8 January 2021) from patients with breast invasive carcinoma to explore the distribution of copy number alterations (CNA) and mutations occurring in the selected genes. The TCGA (Cell 2015 database, *n* = 817) and METABRIC (*n* = 2509) datasets were interrogated, retrieving a total pool of 3326 patients with mutational data, 2989 patients with CNA data, and 2989 with both.

Transcription levels of the selected genes in breast tumors versus health controls were analyzed using the GEPIA2 web server (Gene Expression Profiling Interactive Analysis; http://gepia2.cancer-pku.cn/, accessed on: 8 January 2021). GEPIA2 is an updated version of GEPIA for analyzing the RNA sequencing expression data of 9736 tumors and 8587 normal samples from the TCGA and the GTEx projects. The differences in mRNA levels across intrinsic subtypes were explored in the bc-GenExMiner v4.5. web tool, which comprises RNA-seq data from 4912 tumor samples from the TCGA, SCAN-B, and GTEx projects (http://bcgenex.centregauducheau.fr/, accessed on: 8 January 2021). The UALCAN web resource (http://ualcan.path.uab.edu/, accessed on: 8 January 2021), which integrates omic data from TCGA and MET500, was used to corroborate transcriptional analysis and also provided information on transcriptional differences across tumor stages and on methylation profiles. Individual patient data from TCGA (Cell 2015 database) was downloaded through cBioPortal to analyze the correlation between CNAs and expression levels.

### 4.3. Outcome Analyses

To evaluate the relationship between the gene amplifications and patient clinical prognosis in terms of relapse-free survival (RFS), the publicly available Kaplan–Meier Plotter Online platform (http://www.kmplot.com, accessed on: 8 January 2021) was used, as described previously [25]. Briefly, this tool is capable to assess the effect of 54k genes (mRNA, miRNA, proteins) on survival in multiple cancer types including BC (*n* = 6234), and using GEO, EGA, and TCGA as primary sources. Patients in the database were separated according to the median gene expression values. Patients above the threshold were labelled as “high” expressing ones, while patients below the threshold were labelled as “low” expressing. The two groups were compared using Cox survival analysis.

To analyze the correlation between mutations and patient clinical outcome, the publicly available Genotype-2-Outcome online platform (http://www.g-2-o.com, accessed on: 8 January 2021) was interrogated, as described in previous studies [27]. Briefly, the database allows the association with prognosis of a specific transcriptomic signature linked to a mutation, by classifying patients according to the mean expression of the top 100 most related genes as a surrogate marker of its mutation status. Gene expression is compared between the mutational-carrying and the wild type population and those genes reaching significance are defined as the mutation signature. The median expression values for different transcripts are used as a cut-off to discriminate “high” and “low” expression cohorts, which are compared using a Cox survival analysis. For both tools, patients were stratified by their BC subtype using the PAM50 criteria.

### 4.4. Co-Occurrence

Co-occurrence analysis for gene alterations was evaluated using the cBioPortal online platform (www.cbioportal.org, accessed on: 8 January 2021). This tool calculates an odds ratio (OR) for each pair of query genes, indicating the likelihood that the alterations for the two genes are co-occurrent in the selected cases, by the application of a Fisher’s exact test (statistical significance *p* < 0.05).

### 4.5. Construction and Analysis of PPI Networks and Functional Annotation

We used the online tool STRING (http://www.string-db.org) to construct interactome maps of amplified genes in all subtypes of BC. The closer the local clustering coefficient is to 1, the more likely it is for the network to form clusters. PPI enrichment *p*-value indicates the statistical significance. Proteins are considered hubs when they have more interactions than the average.

### 4.6. Search for Predictive Biomarkers

To evaluate the relationship between gene amplifications and response to a specific therapy, the publicly-available ROC Plotter online tool (http://www.rocplot.org/, accessed on: 8 January 2021) was used, as described previously [26]. Briefly, this tool is capable to link gene expression and response to therapy using transcriptomic data from 3104 breast tumor samples and using GEO, EGA, and TCGA as primary sources. The effect of each gene was interrogated in the (neo)adjuvant setting with endocrine therapy or chemotherapy across the different intrinsic subtypes. A box plot displaying expression levels in responders and non-responders (pCR or 5 years RFS) is provided along with the area under the curve (AUC) and their respective *p*-values. Patients were stratified by their BC subtype using the PAM50 criteria.

### 4.7. Functional Characterization of Mutations

To evaluate the functional effects of the mutations found in the TCGA dataset, several publicly-available online predictor tools were consulted, including SIFT (https://sift.bii.a-star.edu.sg/, accessed on: 8 January 2021; PolyPhen (http://genetics.bwh.harvard.edu, accessed on: 8 January 2021), MutationTester (http://www.mutationtaster.org/, accessed on: 8 January 2021), and FATHMM-MKL (http://fathmm.biocompute.org.uk/, accessed on: 8 January 2021). MaxEntScan was used to estimate if mutations occurring in a splicing site impact the site function (http://hollywood.mit.edu/burgelab/ maxent/Xmaxentscan_scoreseq.html, accessed on: 8 January 2021).

### 4.8. Analysis of Breast Cancer Cell Lines

We interrogated the Cancer Dependency Map (DepMap) portal (https://depmap.org/portal/, accessed on: 8 January 2021), which comprises genomic profiles of hundreds of cancer cell line models and their sensitivity to genetic and small molecule perturbations, and also analyzed individual cell line mutational and transcriptomic data available at the Catalogue of Somatic Mutations in Cancer (COSMIC; https://cancer.sanger.ac.uk/cell_lines, accessed on: 8 January 2021) and at the Cancer Cell Line Encyclopedia (CCLE; https://portals.broadinstitute.org/ccle, accessed on: 8 January 2021). The categorization of cell lines within each intrinsic subtype was performed according to the studies of Dai et al. and Kao et al. and transcriptional data [30,31]. The predictive capability of the set of gene amplifications was explored using data from up to four comprehensive online repositories including Cancer Target Discovery and Development (CTD), Genomics of Drug Sensitivity in Cancer (GDSC1 and GDSC2), and Profiling Relative Inhibition Simultaneously in Mixtures (PRISM), as available in the DepMap portal. Only those with data from at least 5 cell lines were selected for analytical purposes.

### 4.9. Graphical Design

Bars, heatmaps, and regression lines, were represented using GraphPad Prism software (version 8, GraphPad Software, San Diego, CA, USA) in terms of absolute counts, relative frequencies, and hazard ratios. Kaplan-Meier curves were produced by specific online tools as previously described (Kaplan-Meier plotter, Genotype2Outcome, ROC plotter). The interaction network was mapped as default by the STRING database analysis.

## 5. Conclusions

In summary, we performed a comprehensive in silico study of the SUN machinery in breast tumors in search of potential targets for drug development and disease biomarkers. By using a wide repertoire of publicly available databases and analytical tools, we described three gene amplifications (*UBE2T*, *UBE2C*, *BIRC5*) with robust prognostic and predictive relevance in luminal and basal-like tumors, and two mutated genes (*USP9X*, *USP7*) whose surrogate transcriptional signatures have prognostic relevance in luminal A, HER2-enriched, and basal-like tumors. 

Aware of the enormous potential, limitations, and biases intrinsic to computational approaches in cancer genomic research, we advocate for the design of analytical pipelines built on an utmost effort to reduce data noise and with the goal of providing early evidence of potential druggable targets and biomarkers for the future benefit of patients.

## Figures and Tables

**Figure 1 cancers-13-00833-f001:**
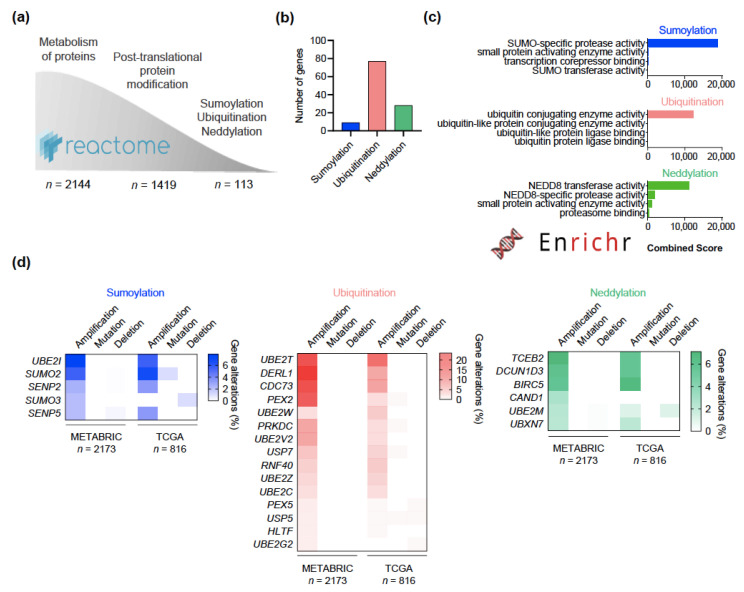
Common genomic alterations in ubiquitination, SUMOylation, and neddylation in breast cancer. (**a**) Flow chart showing the interrogation of the Reactome database to identify genes involved in post-translational protein modifications, particularly ubiquitination, SUMOylation, and neddylation. (**b**) Bar graph summarizing the number of genes participating in each process. Only *UCHL3* participated in both ubiquitination and neddylation. (**c**) Bar graph showing the GO molecular function of the genes previously identified for each process, sorted by the combined score in the EnrichR database. (**d**) Heatmap presenting the frequency of patients from the TCGA and METABRIC datasets with amplifications, mutations, and deletions, of the genes involved in each process. Only genes with a frequency of amplifications higher than 2% are displayed, those under 2% can be found in Appendix A.

**Figure 2 cancers-13-00833-f002:**
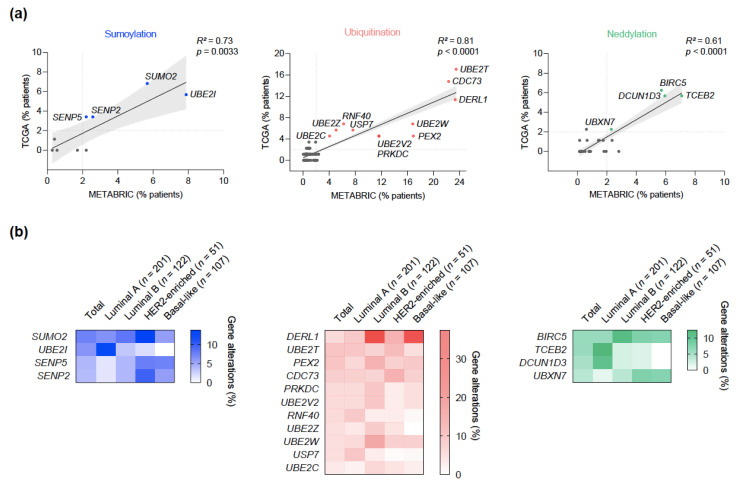
Filtering of gene amplifications. (**a**) Regression analysis of amplification frequencies in the TCGA and METABRIC datasets, highlighting gene amplifications that occur in more than 2% of patients in both. (**b**) Distribution of the frequency of the selected amplified genes across intrinsic subtypes in patients from the TCGA dataset. Population is limited to those patients in which the intrinsic subtype is available in cBioPortal for stratification.

**Figure 3 cancers-13-00833-f003:**
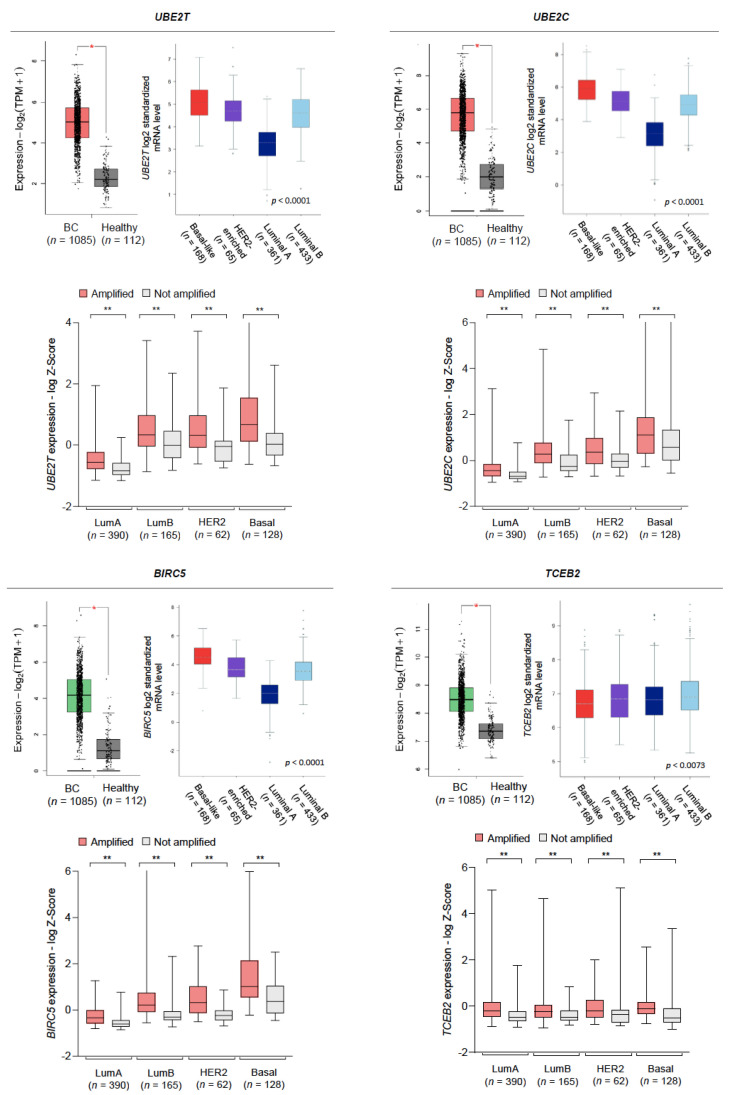
Expression levels of the most frequently amplified genes. Statistically significant differences in gene expression between tumor and healthy samples, along with the distribution of expression levels across intrinsic subtypes (**upper part**). Association of the amplification status with expression levels across intrinsic subtypes (**lower part**). Single asterisk denotes *p* < 0.05 and double asterisk denotes *p* < 0.01 using the Mann-Whitney test.

**Figure 4 cancers-13-00833-f004:**
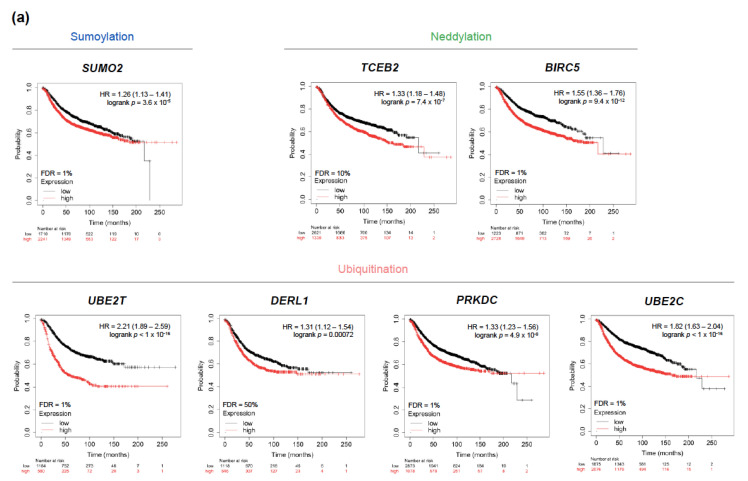
Association between gene expression and a worse relapse-free survival in breast cancer patients. (**a**) Survival analysis for individual genes correlated with poor relapse-free survival among the frequently amplified genes. (**b**) Co-occurrence of mutations in the analyzed population calculated by the odds ratio method. (**c**) Protein-protein interaction map displaying the significant functional network integrated by the selected genes. Line thickness denotes the strength of the association.

**Figure 5 cancers-13-00833-f005:**
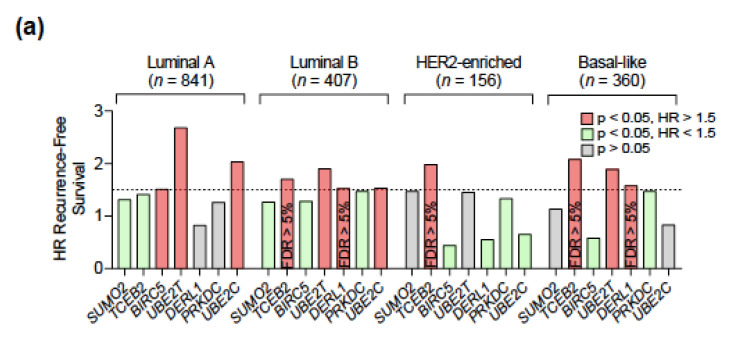
Association between gene expression and a worse relapse-free survival across intrinsic subtypes of breast cancer. (**a**) Filtering of gene amplifications with prognostic capability using the following criteria: statistical significance *p* < 0.05, FDR < 5%, HR of RFS > 1.5. (**b**) Survival analysis for the selected genes correlated with poor relapse-free survival across different intrinsic subtypes of breast cancer.

**Figure 6 cancers-13-00833-f006:**
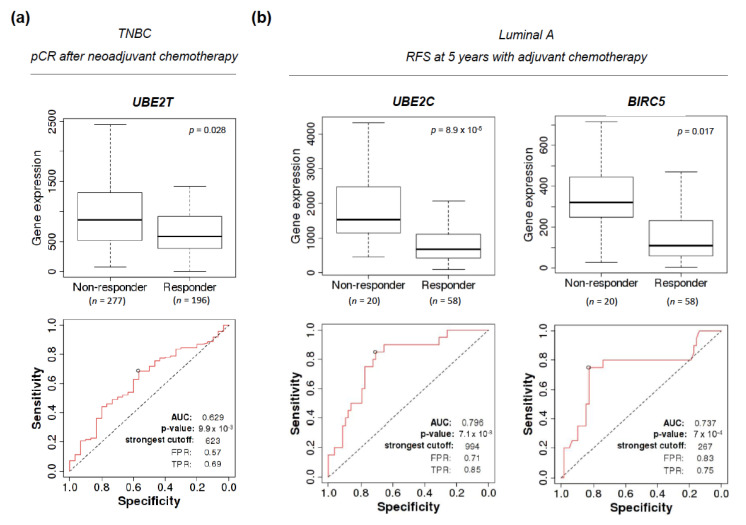
Association between gene expression and response to anti-tumor therapies. (**a**) Difference in gene expression of *UBE2T* in TNBC patients according to the achievement of a pathological complete response (pCR) after neoadjuvant therapy. ROC curve displaying the predictive capability of pCR of *UBE2T*. (**b**) Difference in gene expression of *UBE2C* and *BIRC5* in luminal A patients according to the achievement of 5 years of RFS after adjuvant chemotherapy. ROC curve displaying the predictive capability of 5 years RFS of *UBE2C* and *BIRC5*. (**c**) Difference in gene expression of *UBE2C* and *BIRC5* in luminal A patients according to the achievement of 5 years of RFS with adjuvant endocrine therapy. ROC curve displaying the predictive capability of 5 years RFS of *UBE2T* and BIRC5. Companion graphs for *ESR1* are included as control and help apprehend the potential clinical relevance of the predictive associations.

**Figure 7 cancers-13-00833-f007:**
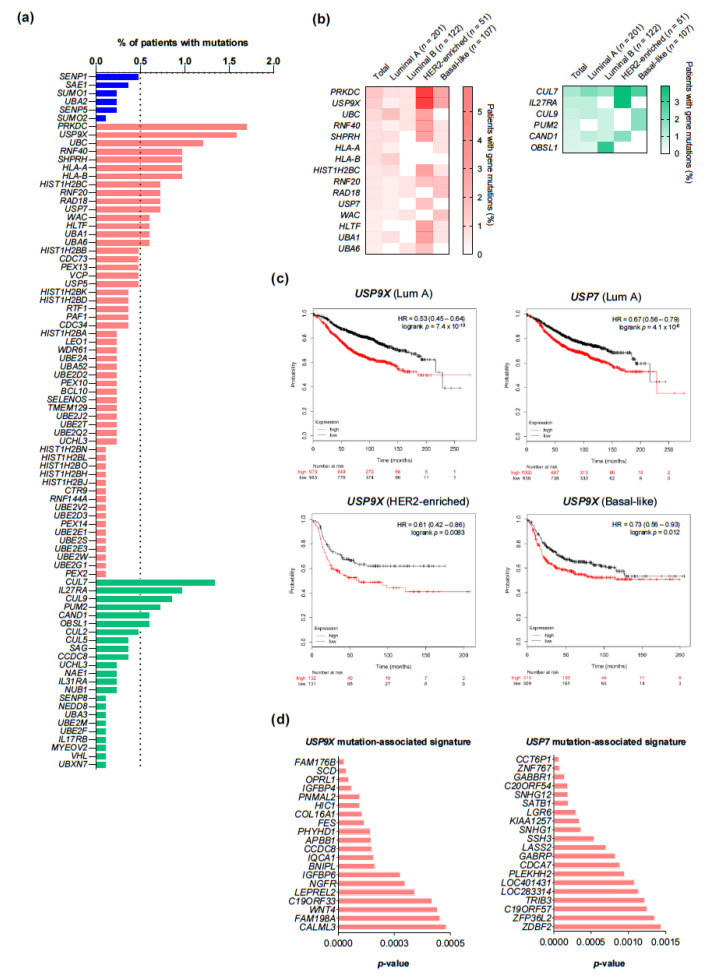
Mutational landscape and prognostic implications. (**a**) Frequency of gene mutations in breast cancer patients from the TCGA dataset. (**b**) Distribution of the frequencies of gene mutations over 0.5% across intrinsic subtypes. Population is limited to those patients in which the intrinsic subtype is available in cBioPortal for stratification. (**c**) Significant associations between the transcriptomic signatures of *USP9X* and *UP7* mutations and overall survival across intrinsic subtypes. (**d**) Top *USP9X* and *USP7*-associated transcriptomic signature components, sorted by *p*-value.

**Figure 8 cancers-13-00833-f008:**
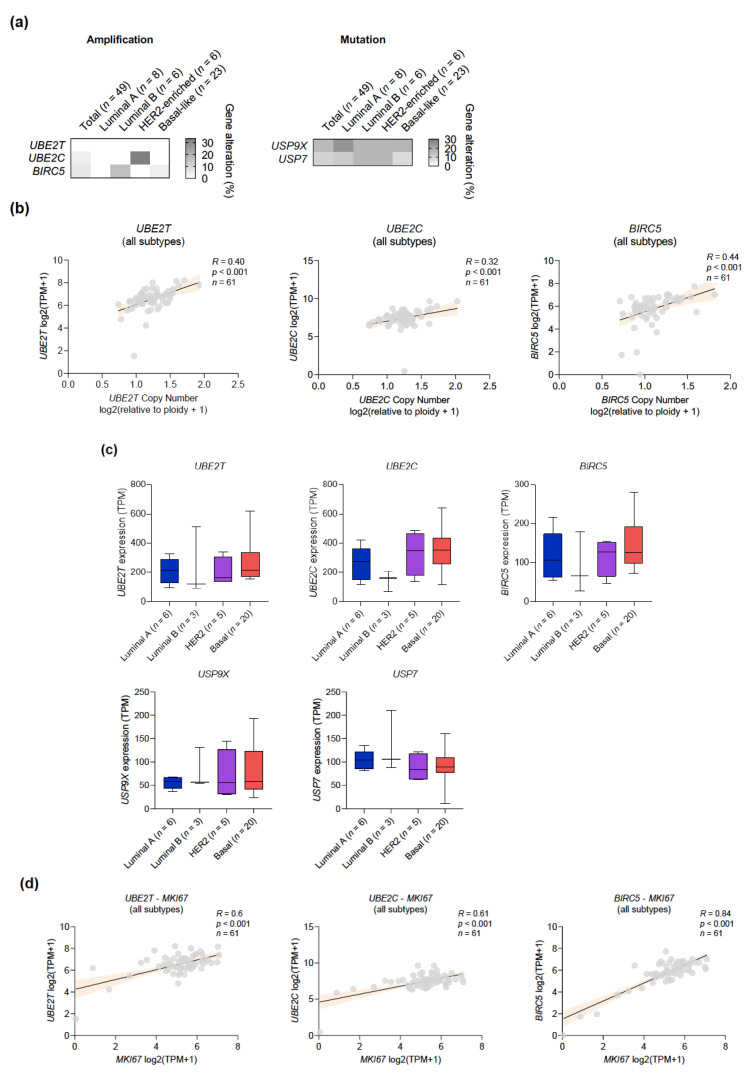
In vitro correlates of the genomic vulnerabilities identified in the SUN machinery in BC. (**a**) Heatmap showing the frequency of *UBE2T*, *UBE2C*, and *BIRC5* amplifications, and *USP9X* and *USP7* mutations, across cell lines of different intrinsic subtypes. (**b**) Positive dependance between copy number and gene expression in BC cell lines. (**c**) Distribution of gene expression across cell lines of the different intrinsic subtypes. (**d**) Positive correlation between the expression levels of *UBE2T*, *UBE2C*, and *BIRC5*, and proliferation marker *MKi67*.

**Table 1 cancers-13-00833-t001:** Summary of genomic vulnerabilities of the SUN machinery with prognostic and predictive implications and potential therapeutic avenues.

Gene (Process)	Alteration	Prognostic Implication	Intrinsic Subtype	Predictive Implication	Intrinsic Subtype	Previous Clinical Evidence	Previous Preclinical Evidence	Potential Therapeutic Avenues	Refs.
*UBE2T* *(Ubi)*	Amp	Poorer survival	Luminal Basal-like	Resistance to neoadjuvant Cx	Basal-like	shorter RFS (basal-like)	Impairment of DNA repair (FANCD2, BRCA1 degradation)	- Targeted inhibition. - Combination strategies (Cx, PARPi)	[36,39]
*UBE2C* *(Ubi)*	Amp	Poorer survival	Luminal	Resistance to adjuvant ET and Cx	Luminal	Shorter RFS and OS (all subtypes)	Resistance to ET, Cx, and RT (mitotic cyclins degradation)	- Targeted inhibition. - Combination strategies (ET, Cx, RT)	[33,34,35,37,38]
*BIRC5* *(Nedd)*	Amp	Poorer survival	Luminal	Resistance to adjuvant ET and Cx	Luminal	shorter RFS/OS (all subtypes) shorter RFS stage II/III (all subtypes)	Apoptosis inhibitor	- Targeted inhibition. - Combination strategies (ET, Cx, PARPi)	[40,41,42,43,44,45]
*USP9X* *(Ubi)*	Mut (deleterious)	Poorer survival	Luminal HER2 Basal-like	-	-	-	- BRCA1 stabilization - Sensitivity to tamoxifen	- Predictor of resistance to ET and sensitivity to Cx and PARPi - Explore actionability in upregulated signature	[46,47,48,49]
*USP7* *(Ubi)*	Mut (deleterious)	Poorer survival	Luminal	-	-	-	Impairment of mitotic progression	- Predictor of sensitivity to Cx - Explore actionability in upregulated signature	[50,51,52]

Ubi: ubiquitination; Nedd; neddylation; Amp: amplification; Mut: mutation; Cx: chemotherapy; ET: endocrine therapy; RFS: relapse-free survival; OS: overall survival; RT: radiotherapy; ER: estrogen receptor.

## Data Availability

The data presented in this study are available within the article and its supplementary material.

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
