# Peer review of "Mapping of Genomic Vulnerabilities in the Post-Translational Ubiquitination, SUMOylation and Neddylation Machinery in Breast Cancer"

_cancers, 2021, doi:10.3390/cancers13040833_

Round 1

Reviewer 1 Report

This topic is very interesting.

However, could you show clinical characteristics of patients as far as possible, if you show RFS.

Author Response

We thank the reviewer for the positive comments on our work

It is our aim to leverage computational approaches in clinically accurate and relevant scenarios. Clinical-pathological features of patients from the TCGA and METABRIC datasets who have available survival data are now summarized in Supplementary Table 1b, which is referenced in ‘Results. 2.3. Amplification of SUN genes predicts an unfavorable outcome in breast tumors’ (please see the added text in blue).

Further detail is not provided due to space constraints and could be included if indicated. The utilized online tools make computation using highly cited data originally published in seminal papers (e.g. TCGA, Nature, 6335 citations; Pereira, Nat Comm, 504 citations)

Reviewer 2 Report

The paper by Fuentes-Antrás et al., titled “Mapping of Genomic Vulnerabilities in the Post- 
translational Ubiquitination, SUMOylation and 
Neddylation Machinery in Breast Cancer” 
explores the genomic alterations occurring in ubiquitination and ubiquitin-like SUMOylation, and neddylationin (collectively termed SUN) in breast cancer by an in silico analysis using data from publicly available repositories.

By analyzing multiple large publicly accessible databases and analytical webs for cancer omics, the authors identified some genes of the SUN components that are amplified in breast cancer. In addition, they found that the mRNA levels of amplified genes tended to be higher in breast tumors than in the healthy state, and that amplification of SUN genes predict an unfavorable outcome in breast tumors. Being the dysregulation of this post-translational machinery involved in tumorigenesis and drug resistance, the aim of this study is to provide a rationale for future drug development but also to propose new potential prognostic markers.

General considerations

The work is very interesting also in view of the development of drugs targeting these pathways that the authors also show to be intersected with hormonal ones.

I believe it is an important work as it focuses on specific pathways not yet sufficiently investigated in the oncology field. It is well organized and written smoothly and the conclusions reached by the authors are supported by the data reported in the work.

Specific points:

Apart from a moderate general linguistic revision, the paper could increase its usefulness for the reader by adding a table summarizing the amplifications / mutations of the specific genes found in the different types of breast cancer specifying which of these could make use of a therapeutic strategy based on drugs targeted towards these pathways.

Author Response

We thank the reviewer for the positive comments on our work.

Key genomic alterations, specific intrinsic subtypes, existing (pre)clinical evidence, and potential clinical implications are now briefly collected in Table 1 so that readers can easily integrate the findings discussed in the text. Table 1 is referenced in the third paragraph of the section ‘Discussion’ (in blue)

Reviewer 3 Report

The authors performed a total in silico analysis of mutated genes of ubiquitination and ubiquitin-like SUMOylation and neddylation pathways in breast to find potential targets for drug development and disease biomarkers.

By using online database, the authors selected three gene amplifications for UBE2T, UBE2C, BIRC5 observing a prognostic and predictive relevance in luminal and basal-like tumors, and two mutated genes (USP9X, USP7) that could be a prognostic power in luminal A, HER2-enriched, and basal-like tumors.

I am quite skeptical with this exclusive use of online tolls without further characterization and validation, at least on a panel of cell lines recapitulating BC subtypes or on an available and well-controlled cohort of patients. Regarding this topic, the works of the group of L. Langerød and AL Børresen-Dale about the classifications in molecular subtypes of BC have made school (Nat Commun. 2016 Sep 26; 7: 12910. Doi: 10.1038 / ncomms12910.; Cell Rep. 2016 Aug 16; 16 (7): 2032-46. Doi: 10.1016 / j.celrep.2016.07.028 .; Nature. 2012 Apr 18; 486 (7403): 346-52. Doi: 10.1038 / nature10983.).

I could not evaluate the supplementary figures and tables, even if I followed the description in the manuscript, as in the supplementary material I found only the word file with the legends to the figures.

I think that Fig.3 and Fig.4 are the least convincing figures in terms of online tools used.

So, I consider the results shown in this manuscript as preliminary. Overall, it is my opinion that this manuscript is not acceptable for publication on Cancers.

Author Response

We thank the reviewer for the constructive criticism of our work and valuable suggestions, and also regret that the supplementary material containing large-cohort validation was not available for review.

It is our aim to leverage computational approaches in clinically accurate and relevant scenarios in order to progress as much as possible without inefficiently compromising patients and resources. The utilized online tools make computation using highly cited data originally published in seminal papers (e.g. TCGA, Nature, 6335 citations; Pereira, Nat Comm, 504 citations; and the works referenced by the reviewer) and the widespread use of these data supports its reliability. Further, some of the authors are responsible for the creation of these analytical tools and have extensive and rigorous knowledge of data sources and filtering procedures (e.g. Nagy A, Sci Rep; Fekete J, Int J Cancer; Pongor L, Genome Med).

Following the suggestions of the reviewer, we conducted a comprehensive in silico characterization of BC cell lines using mutational, transcriptional, and sensitivity data, in search of potential in vitro models for the identified genomic alterations. This data was extracted from core web repositories including the Cancer Dependency Map Portal (depmap; Broad Institute, Wellcome Sanger Institute), the Catalogue of Somatic Mutations in Cancer (COSMIC; Wellcome Sanger Institute), and the Cancer Cell Line Encyclopedia (CCLE; Broad Institute, Novartis Institutes for Biomedical Research). Please see the added text (in blue) in the new section ‘Results. 2.7. Exploring genomic vulnerabilities in a large panel of breast cancer cell lines’, along with ‘Figure 8. In vitro correlates of the genomic vulnerabilities identified in the SUN machinery in BC’ and ‘Materials and Methods. 4.8. Analysis of breast cancer cell lines’. Further detail on the distribution of gene amplifications/mutations across the most frequently utilized breast cancer cell lines, and sensitivity data, are provided in Suppl Table 4 and Suppl Figures 4 and 5.

Round 2

Reviewer 3 Report

The revision version of the manuscript is more satisfactory than the original. Even the supplementary data that I was able to view adequately support the results. This version is suitable for publication.